# Patient-centred outcomes following non-operative treatment or appendicectomy for uncomplicated acute appendicitis in children

Nigel J Hall ![ORCID],[1,2] Frances C Sherratt ![ORCID],[3] Simon Eaton ![ORCID],[4] Erin Walker,[5] Maria Chorozoglou ![ORCID],[6] Lucy Beasant,[7] Michael Stanton,[2] Harriet Corbett,[8] Dean Rex,[9] Natalie Hutchings,[10] Elizabeth Dixon,[10] Esther Crawley ![ORCID],[7] Jane Blazeby,[11] Bridget Young,[3] Isabel Reading[12]

## ABSTRACT

While non-operative treatment has emerged as an alternative to surgery for the treatment of uncomplicated acute appendicitis in children, comparative patient-centred outcomes are not well documented. We investigated these in a feasibility randomised trial. Of 57 randomised participants, data were available for 26. Compared with appendicectomy, children allocated to non-operative treatment reported higher short-term quality of life scores, shorter duration of requiring analgesia, more rapid return to normal activities and shorter parental absence from work. These preliminary data suggest differences exist in recovery profile and quality of life between these treatments that are important to measure in a larger RCT. Trial registration number is ISRCTN15830435.

Treatment of appendicitis in children has traditionally been by appendicectomy. Over recent years, there has been increasing interest in the use of non-operative treatment as an alternative to appendicectomy[1 2] but comparative outcomes of these two different treatment modalities are poorly understood. Previous reports have tended to focus on outcomes most relevant to surgeons such as surgical complications and other treatments required rather than considering a full range of outcomes that we know are important to patients and families.[3] Here, we report a number of patient-centred outcomes including health-related quality of life (HRQoL) in data arising from a feasibility randomised controlled trial (RCT).

A feasibility RCT comparing appendicectomy and non-operative treatment in children with suspected uncomplicated acute appendicitis was undertaken in three UK centres over a 12-month period beginning March 2017. Full details of the methodology, key clinical findings and feasibility findings

of the trial have been previously reported.[4] To understand patient-centred markers of recovery, we used parent completed diary cards following hospital discharge to record duration of analgesia use, time taken to return to normal and full activities and duration of parental absence from work. We used the CHU-9D tool[5] to measure HRQoL during the trial period. Since this was a feasibility trial, we report descriptive data without formal statistical analyses. Data are reported on an intention-to-treat basis as per trial allocation.

Patients and families were involved in the design of the trial, development of patient and family facing materials and defining outcomes to be measured.

A total of 57 participants were enrolled in the trial of whom 28 were allocated to appendicectomy and 29 to non-operative treatment. Data relating to early post-discharge recovery were available for 26 participants—15 in the appendicectomy arm and 11 in the non-operative treatment arm (Table 1). The proportion of participants taking analgesia was generally lower for each day following discharge in the non-operative treatment arm than the appendicectomy arm. Following non-operative treatment, participants were able to return to normal or full activities earlier than those in the appendicectomy arm. Fewer than half of participants in the appendicectomy reported being able to participate in full activities by 2 weeks following discharge. Parents of participants in the non-operative treatment arm were able to return to work earlier than those in the appendicectomy arm. HRQoL assessed using the CHU-9D tool at enrolment, hospital discharge and further time points during the 6-month follow-up period is shown in figure 1.

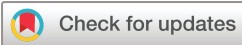

For numbered affiliations see end of article.

**Correspondence to**
Nigel J Hall; n.j.hall@soton.ac.uk

While we acknowledge a relatively small sample size and incomplete data, our observation of an apparent difference in early post-discharge recovery profile between participants allocated to appendicectomy or non-operative treatment is an important finding with relevance for both clinical practice and future research. Avoidance of general anaesthesia, trauma of surgery and lack of exposure to operative/postoperative complications are viewed as potential benefits of non-operative treatment over surgery by patients and parents. Our data suggest there may be additional benefits to non-operative treatment over surgery in terms of more rapid return to baseline activity status and societal benefits with fewer days of parental work absence. Patients and parents may wish to consider these as well as more direct clinical outcomes when considering choice of treatment. Furthermore, there appear to be short-term differences in HRQoL between treatment arms, which may be clinically meaningful and will inform the design of future cost–utility analyses.

Overall our data emphasise the importance of measuring a full range of outcomes including these patient-centred outcomes in future research and provide greater insights into short-term differences in outcomes between treatment arms.

**Author affiliations**
[1]University Surgery Unit, Faculty of Medicine, University of Southampton, Southampton, UK
[2]Department of Paediatric Surgery and Urology, Southampton Children's Hospital, Southampton, UK
[3]Department of Public Health, Policy and Systems, University of Liverpool, Liverpool, UK
[4]UCL Great Ormond Street Institute of Child Health, London, UK
[5]Great Ormond Street Hospital For Children NHS Trust, London, UK
[6]Southampton Health Technology Assessment Centre, University of Southampton Faculty of Medicine, Southampton, UK
[7]Centre for Academic Child Health, Bristol Medical School, University of Bristol, Bristol, UK
[8]Department of Surgery, Alder Hey Children's NHS Foundation Trust, Liverpool, UK
[9]Paediatric Surgery, St George's University Hospitals NHS Foundation Trust, London, UK
[10]Southampton Clinical Trials Unit, Faculty of Medicine, University of Southampton, Southampton, UK
[11]Bristol and Weston Biomedical Research Centre, Population Health Sciences, University of Bristol, Bristol, UK
[12]Primary Care, Population Sciences and Medical Education, Faculty of Medicine, University of Southampton, Southampton, UK

**Contributors** Study design: NJH, FCS, SE, EW, MC, LB, NH, ED, EC, JB, BY and IR. Study delivery: NJH, FCS, MS, HC, DR, NH, ED, BY and IR. Data collection: NJH, MS, HC, DR, NH and ED. Data analysis: NJH, MC and IR. Manuscript writing: NJH, NH, ED, MC and IR. Review of manuscript: FCS, SE, EW, MC, LB, MS, HC, DR, EC, JB and BY.

**Funding** The study was funded by the National Institute for Health Research Health Technology Assessment Program (ref: 14/192/90).

**Disclaimer** The views expressed are those of the authors and not necessarily those of the NHS, the NIHR or the Department of Health. The funders had no role in developing the protocol.

**Competing interests** No, there are no competing interests.

**Patient consent for publication** Not applicable.

**Table 1** Profile of post-discharge recovery in each treatment arm

| Day following discharge | Analgesia used | | Able to do normal activities | | Able to do full activities | | Parental work absence | |
|---|---|---|---|---|---|---|---|---|
| | NOT | APP | NOT | APP | NOT | APP | NOT | APP |
| 0 | 27 | 80 | 82 | 7 | 36 | 0 | 36 | 67 |
| 1 | 27 | 73 | 91 | 7 | 36 | 7 | 36 | 53 |
| 2 | 18 | 60 | 91 | 33 | 45 | 7 | 9 | 40 |
| 3 | 18 | 60 | 91 | 47 | 54 | 13 | 9 | 40 |
| 4 | 27 | 40 | 100 | 53 | 73 | 20 | 9 | 40 |
| 5 | 20 | 47 | 90 | 67 | 80 | 20 | 10 | 27 |
| 6 | 20 | 20 | 90 | 73 | 80 | 20 | 10 | 27 |
| 7 | 20 | 27 | 100 | 80 | 90 | 20 | 20 | 27 |
| 8 | 10 | 13 | 100 | 80 | 80 | 27 | 10 | 20 |
| 9 | 10 | 13 | 100 | 87 | 90 | 33 | 0 | 20 |
| 10 | 10 | 7 | 80 | 93 | 70 | 40 | 0 | 13 |
| 11 | 10 | 7 | 90 | 93 | 90 | 40 | 0 | 13 |
| 12 | 10 | 13 | 90 | 87 | 90 | 40 | 0 | 7 |
| 13 | 0 | 7 | 90 | 87 | 90 | 47 | 0 | 7 |
| 14 | 0 | 13 | 70 | 87 | 70 | 47 | 0 | 20 |

Data are percentage of participants reporting data on each day. For all outcomes, n=15 for APP arm; for NOT arm n=11 for days 0–4 and n=10 for days 5–14.
APP, appendicectomy arm; NOT, non-operative treatment arm.

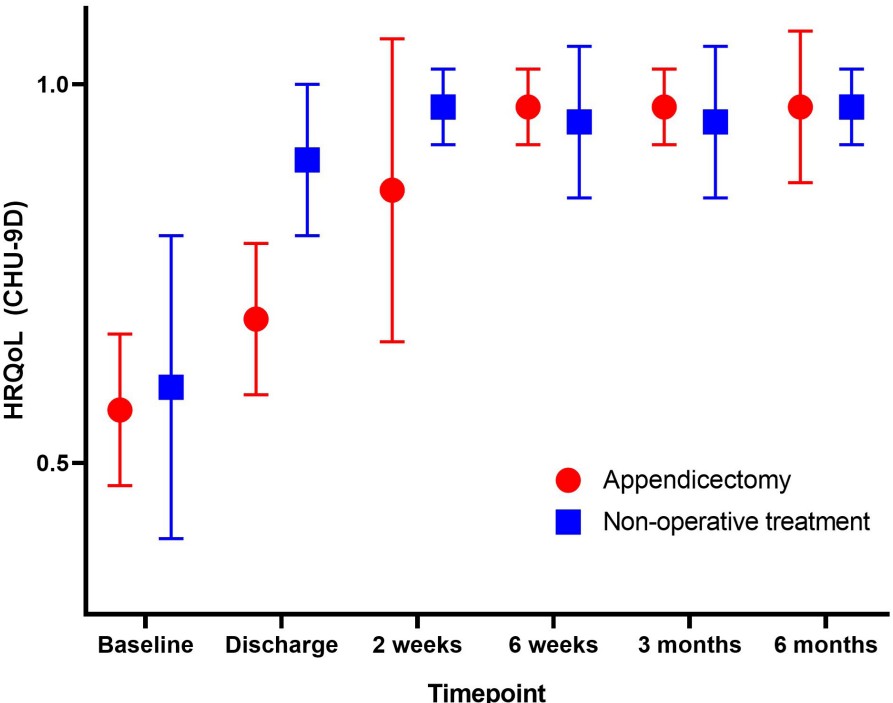

**Figure 1** Health-related quality of life (HRQoL) (CHU-9D) during the trial. Data are mean with SD, HRQoL is scored on a scale from 0 (dead) to 1 (perfect health).

**Ethics approval** This study involves human participants and was approved by Hampshire A Research Ethics Committee (ref 16/SC/0596). Participants gave informed consent to participate in the study before taking part.

**Provenance and peer review** Not commissioned; externally peer reviewed.

**ORCID iDs**
Nigel J Hall http://orcid.org/0000-0001-8570-9374
Frances C Sherratt http://orcid.org/0000-0003-4147-9305
Simon Eaton http://orcid.org/0000-0003-0892-9204
Maria Chorozoglou http://orcid.org/0000-0001-5070-4653
Esther Crawley http://orcid.org/0000-0002-2521-0747

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
