## [Reviewer comments · BMJ Paediatrics Open]

ARTICLE DETAILS

TITLE (PROVISIONAL)	
AUTHORS	

VERSION 1 – REVIEW

REVIEWER	Reviewer name: Dr. Peter Flom Institution and Country: Peter Flom Consulting, 515 West End Ave New York, 10024, United States Competing interests: None
REVIEW RETURNED	15-Sep-2022

GENERAL COMMENTS	I confine my remarks to statistical aspects of this paper. These were quite straightforward and, given the purpose of the paper, appropriate. Peter Flom
---

REVIEWER	Reviewer name: Ram iNataraja Institution and Country: United Kingdom of Great Britain and Northern Ireland Competing interests: None
REVIEW RETURNED	13-Oct-2022

GENERAL COMMENTS	This is a well written feasibility study for the investigation of the non-operative treatment of uncomplicated appendicitis in children. It is well designed and executed. I have a few queries about the study: There seems to be a low rate of uncomplicated appendicitis in these 3 tertiary centres despite an appendectomy being one of the most common operations that we perform. 57 were recruited which was described as 44% of eligible patients who were approached and recruited. This would roughly equal 130 patients or 43 per centre/yr. What is the actual incidence of uncomplicated appendicitis in each centre as it may appear that quite a few were not approached for recruitment at all? There was a relatively high complicated appendicitis rate in both arms of the trial but less than a 1/3 of patients received investigation pre-operatively with an USS. This is becoming routine practice in many centres hence would the inclusion of this pre-operative investigation improve the trial? Certainly there are USS characteristics that indicate completed appendicitis and also the presence of a faecolith that has been shown as a likely factor for failure of non-operative treatment. I appreciate that this may not be practically possible in all centres.
---

	The complications of the disease and treatment is a secondary trial outcome. However neither wound infections or intra-abdominal abscesses are mentioned in the trial. Although IAAs should be relatively rare, a non-operative participant did represent with an appendix mass so this should be reported. I would expect there to a wound infection incidence with both laparoscopic and open appendicectomies included. There has also been a pilot RCT in children in Sweden with 50 patients so is the justification for this feasibility trial the geographical location of the patients? There is also a large number of RCTs in adult populations since 1997 so the technique is not novel. This should be discussed. Apologies if I have misinterpreted the differences between a pilot and a feasibility trial, as my impression was a feasibility trial was to test components or techniques that had not been applied yet in a clinical setting prior to conducting an adequately powered trial? These queries do not detract from the importance of this trial once conducted as this is a very topical issue at the moment in paediatric surgery and the authors should be congratulated on performing a multi-centre RCT.
--	---

REVIEWER	Reviewer name: Jason Fisher Institution and Country: United Kingdom of Great Britain and Northern Ireland Competing interests: None
REVIEW RETURNED	19-Oct-2022

GENERAL COMMENTS	The authors have presented a small RCT regarding the nonoperative treatment of uncomplicated appendicitis in children. There is a growing body of literature on this topic, including several RCT and non-RCT studies over the past decade. Reporting of a diverse array of experiences in this management algorithm is critical to build a robust repository of evidence on which to guide pediatric surgical care. Indeed, I would point the authors to our own RCT published earlier this year (https://pubmed.ncbi.nlm.nih.gov/34674843/) on this exact topic -- I reference this certainly not to suggest it be included as a reference, but that it is a similarly small study that attempted to address similar issues studied here and perhaps the authors may find it helpful. Regarding this manuscript, I have several questions/comments that should be addressed prior to publication.  1. The authors emphasize repeatedly that this is a "feasibility" study and that there is limited attention paid in current trials to patient-centered outcomes. "Feasibility" of this approach has now been long-established, and the challenges of randomization particularly for the surgical care of children have also been broadly discussed in the literature. Thus, I would encourage the authors to uncouple the idea of "feasibility" from this study, as it is not in-tune with the current state of this topic. Sure, "larger trials" are always needed, but that doesn't relegate smaller ones to being simply for "feasibility". 2. The authors need to better clarify their recruitment numbers. The eligibility for this study appears to allow flexibility in recruitment (compared to many published trials which have more stringent criteria for duration of symptoms, WBC, and appendiceal size). This appears to have been done intentionally to allow for
---

	"replication of practice patterns" and delivery of a "pragmatic" trial as described by the authors. That is quite reasonable, but with such loose inclusion criteria, it creates the potential for incredible bias in patient selection, in that we don't have the denominator as to how many patients were truly eligible. It appears that only 114 patients were approached to enroll (authors report 57 enrolled corresponding to a 50% enrollment rate). I find it hard to believe that across 3 large UK centers over 12 months, only 114 patients presented with uncomplicated appendicitis, meaning that many more were likely eligible but not approached for enrollment... why? And the "why" is actually less important than just reporting what the total eligibility was. Indeed, recruitment into these studies is incredibly challenging. In fact, 50% enrollment is extremely high compared to published studies (our enrollment rate was only 16%) -- thus, it suggests ALL eligible patients were likely not approached, and thus selection bias looms large. I would recommend inclusion of a flow chart of patients which would provide clarity here, starting at eligibility (something like Figure 1 in our referenced paper). 3. The reported complication rate for the operative appendicitis cases seems unusually high (30%) for uncomplicated appendicitis, inclusive of multiple readmissions. The authors should provide more details on what these complications were. This certainly will make the nonoperative treatment pathway seem more favorable against an elevated surgical complication rate. Again, this may be the result of "non-stringent" inclusion criteria. It appears that 30% of the presumably uncomplicated appendicitis patients in the operative group actually had perforated appendicitis! These patients never should have been enrolled, and certainly would drive the complication rate higher in an otherwise uncomplicated cohort. This further speaks to the sub-optimal inclusion/exclusion strategy. The authors state diagnostic imaging is not always utilized -- I would counter and say that if you are considering nonoperative management, you want to be as certain as possible that this truly is uncomplicated appendicitis -- and imaging can certainly help that -- otherwise you are setting the antibiotics-only patients up to fail. 4. Similarly, the true success rate of the nonoperative arm was only 44%. The authors report 70% in the nonop group were discharged home, but then 9 were readmitted (7 w/ recurrent appendicitis) -- that math yields 44% success, which is quite low -- now several who failed (n=4 i think) were also found to have perforated appendicitis, again revealing that these patients never should have been included in the study. 5. While the authors' findings of the improved patient-centered outcomes in the nonop group are consistent with other published studies, post-discharge data were only available in 26/57 participants -- this is less than 50%, and likely insufficient to provide valid conclusions on the patient-centered outcomes. These limitations are not necessarily disqualifying for publication, but need to be proactively addressed in the methods and discussion sections.
--	--

REVIEWER	Reviewer name: Dr. Erik Skarsgard Institution and Country: BC Children's Hospital, Surgery
-----------------	---

	K0-110 ACB, 4480 Oak Street, Vancouver, Canada Competing interests: None
REVIEW RETURNED	19-Oct-2022

GENERAL COMMENTS	This manuscript is from a feasibility RCT comparing outcomes of non-operative treatment vs appendectomy for simple acute appendicitis in children. The trial's primary outcome (recruitment rate), as well as details of secondary, 6-month clinical outcomes have been published previously (Hall NJ, Eaton S, Sherratt FC, et al. CONservative TRreatment of Appendicitis in Children: a randomised controlled feasibility Trial (CONTRACT). Arch Dis Child. 2021 Jan 13;106(8):764–73). These outcomes include initial success rate of non-operative treatment, rate of recurrent appendicitis and overall rate of avoidance of appendectomy in the non-operative group; and in the appendectomy group, histologic findings in the resected appendix, and postoperative complications and readmissions. This manuscript also includes a detailed tabulation of adverse events, severity assignment and actions taken for both groups. The current manuscript cites this publication (reference 2) and acknowledges the previous reporting of recruitment rate “and other feasibility outcomes”. In the Methods of the current manuscript, a bulleted list of outcomes is provided under the Heading of “Secondary Trial Outcomes reported here...”. These include “safety and overall success of initial non-operative treatment”, “complications of disease and treatment” and “rate of recurrent appendicitis during 6-month followup period”. Although not explicitly stated, it is implied that these outcomes have not been previously reported, however, after reading reference 2, I believe that these outcomes have been published. The remaining outcomes (length of stay, profile of discharge recovery and health-related QOL are new. While I accept that it is reasonable, and often “good science” to publish multiple articles from a single study dataset, it is the authors’ responsibility to ensure avoidance of redundancy in these publications, or if it is unavoidable, to explicitly acknowledge it-- and I don’t think that has happened here. The remainder of my comments are in reference to the outcomes that have not been previously reported. 1. Length of hospital stay is reported in table 2 and includes time measurements from randomization to decision to discharge and actual discharge. No explanation is provided to explain the process from decision to actual discharge, but it is presumed that this would be agnostic to the treatment arm. Unless some additional explanation is provided, I'd suggest that only time to actual discharge be presented, since this is the outcome that matters from the patient/family perspective. 2. The outcomes reported by the patient diary cards include duration of analgesia use, return to normal/full activities and parental absence from work, and are based on the return of these cards by families. Not only was the overall rate of return low, it was also disparate between groups (11/29=38% in the non-operative group and 16/28 = 57% in the operative group). I think that the results need to be interpreted in the context of the low and unequal respondent rates of the groups. Other than the general
---

	acknowledgement of study limitations of a small sample size and short duration of followup, there is no mention of the possibility of contribution of sampling bias in the family's decision to complete or return the diary cards. I accept that this is a "feasibility" trial which justifies the small sample size, but I still think recognition and discussion of these challenges is important, since they will require mitigation in the full trial. 3. Quality of life was assessed at multiple time points using the CHU-9D tool. No description of the tool itself, to whom or how it was administered (patient or parent), and evidence of its previous validation in childhood appendicitis populations is provided. This should be included in the methods.
REVIEWER	Reviewer name: Mrs. Coral Smith Institution and Country: University of Nottingham, United Kingdom of Great Britain and Northern Ireland Competing interests: None
REVIEW RETURNED	21-Oct-2022
GENERAL COMMENTS	I really enjoyed the content of your publication, It was an interesting article which contributed to an under researched area. These are my comments for your consideration. Page 2, the last sentence of the background under abstract is too long without punctuation. Page 3 line 25 consider a comma after surgeons. First use of RCT I cant see you spelt it out (not sure if you need to as a well recognized abbreviation). Page 3, Under introduction line 31/32 cross off one of the "ins". Page 3 under Methods line 49/50 consider a comma after March 2017. Page 5, non operative treatment arm - line 5 consider changing "are" to were. and I naturally put a pause after the word operation in the same sentence so consider a comma there. Lastly page 14, line 50 measuring is spelt wrong.

VERSION 1 – AUTHOR RESPONSE

Patient centred outcomes following non-operative treatment or appendicectomy for uncomplicated acute appendicitis in children – results of a feasibility randomised controlled trial."

Following review of your article to BMJ Paediatrics Open, we invite you to submit a major revision as a research letter (600 words max +100 word unstructured abstract, 2 figures/tables, max 6 references).

Dear Dr. Brodlie and Professor Choonara

Editor in Chief Comments to Author :

We discussed whether we should reject outright, but felt it would be better to offer a research letter. We agreed with reviewer 4 that much of the paper repeats information in reference 2, eg Table 1 and Methods. As n=27, you need to be cautious in your conclusions. Focus on Fig 4. Fig 3 would probably be better as a table with actual numbers and summary data, rather than results for each day.

We acknowledge that much of the methods section and some of the results section was repeating information previously published (and acknowledged), however we felt that this was important to

include in the previous version of a full article. We have now removed this, simply referencing our previous publication in a shorter research letter.

Title shorten to "Patient centred outcomes following non-operative treatment or appendicectomy for uncomplicated acute appendicitis in children"

This has been done

Associate Editor

Comments to the Author:

Thank you for submitting your manuscript to BMJPO.

It has been peer reviewed and considered at an editorial level.

Importantly, a reviewer has noted that it appears that some of the data have been published from this study previously in:

Hall NJ, Eaton S, Sherratt FC, Reading I, Walker E, Chorozoglou M, Beasant L, Wood W, Stanton M, Corbett H, Rex D, Hutchings N, Dixon E, Grist S, Crawley EM, Young B, Blazeby JM. CONservative TRreatment of Appendicitis in Children: a randomised controlled feasibility Trial (CONTRACT). Arch Dis Child. 2021 Jan 13;106(8):764–73. doi: 10.1136/archdischild-2020-320746. Epub ahead of print. Erratum in: Arch Dis Child. 2021 Nov;106(11):e43. PMID: 33441315; PMCID: PMC8311091.

Although the QoL data were not included in this.

We agree and had already knowledge this in our previous submission. We have now limited our report to new data only and referenced our previous publication for both methods and other trial outcomes.

Please also note the other reviewers comments.

We would like to offer the opportunity to resubmit as a research letter that focuses on the outcomes not previously reported and responds to the various reviewers comments fully.

We have pleasure in providing a research letter for your further consideration and are grateful for this opportunity.

Reviewer: 1

Dr. Peter Flom, Peter Flom Consulting

I confine my remarks to statistical aspects of this paper.

These were quite straightforward and, given the purpose of the paper, appropriate.

Thank you

Reviewer: 2

Ram iNataraja

This is a well written feasibility study for the investigation of the non-operative treatment of uncomplicated appendicitis in children. It is well designed and executed.

I have a few queries about the study:

There seems to be a low rate of uncomplicated appendicitis in these 3 tertiary centres despite an appendicectomy being one of the most common operations that we perform. 57 were recruited which was described as 44% of eligible patients who were approached and recruited. This would roughly equal 130 patients or 43 per centre/yr. What is the actual incidence of uncomplicated appendicitis in each centre as it may appear that quite a few were not approached for recruitment at all?

We have described recruitment previously in our main feasibility trial publication.

There was a relatively high complicated appendicitis rate in both arms of the trial but less than a 1/3 of patients received investigation pre-operatively with an USS. This is becoming routine practice in many centres hence would the inclusion of this pre-operative investigation improve the trial? Certainly there are USS characteristics that indicate completed appendicitis and also the presence of a faecolith that has been shown as a likely factor for failure of non-operative treatment. I appreciate that this may not be practically possible in all centres.

The complications of the disease and treatment is a secondary trial outcome. However neither wound infections or intra-abdominal abscesses are mentioned in the trial. Although IAAs should be relatively rare, a non-operative participant did represent with an appendix mass so this should be reported. I would expect there to a wound infection incidence with both laparoscopic and open appendicectomies included.

Key outcomes you request are included in our previous publication and therefore not included here.

There has also been a pilot RCT in children in Sweden with 50 patients so is the justification for this feasibility trial the geographical location of the patients? There is also a large number of RCTs in adult populations since 1997 so the technique is not novel. This should be discussed. Apologies if I have misinterpreted the differences between a pilot and a feasibility trial, as my impression was a feasibility trial was to test components or techniques that had not been applied yet in a clinical setting prior to conducting an adequately powered trial?

The feasibility aspect of this trial was to assess the feasibility of delivering a future trial rather than the feasibility of the intervention working. Again this is highlighted and discussed in our previous report.

These queries do not detract from the importance of this trial once conducted as this is a very topical issue at the moment in paediatric surgery and the authors should be congratulated on performing a multi-centre RCT.

Reviewer: 3

Jason Fisher

Comments to the Author

The authors have presented a small RCT regarding the nonoperative treatment of uncomplicated appendicitis in children. There is a growing body of literature on this topic, including several RCT and non-RCT studies over the past decade. Reporting of a diverse array of experiences in this management algorithm is critical to build a robust repository of evidence on which to guide pediatric surgical care. Indeed, I would point the authors to our own RCT published earlier this

year

(<https://eur03.safelinks.protection.outlook.com/?url=https%3A%2F%2Fpubmed.ncbi.nlm.nih.gov%2F34674843%2F&data=05%7C01%7Cn.j.hall%40soton.ac.uk%7Cf04505a816e14ec6365a08dab378d825%7C4a5378f929f44d3ebe89669d03ada9d8%7C0%7C0%7C638019628848371517%7CUnknown%7CTWFpbGZsb3d8eyJWljiMC4wLjAwMDAiLCJQIjoiV2luMzliLCJBTiI6IjEhaWwiLCJXVCi6Mn0%3D%7C3000%7C%7C&sddata=AZjNP9yrmSsfntEBFdnxdE%2BF2YCRPZSVI6ms1K3Np14%3D∓reserved=0>) on this exact topic -- I reference this certainly not to suggest it be included as a

reference, but that it is a similarly small study that attempted to address similar issues studied here and perhaps the authors may find it helpful. Regarding this manuscript, I have several questions/comments that should be addressed prior to publication.

1. The authors emphasize repeatedly that this is a "feasibility" study and that there is limited attention paid in current trials to patient-centered outcomes. "Feasibility" of this approach has now been long-established, and the challenges of randomization particularly for the surgical care of children have also been broadly discussed in the literature. Thus, I would encourage the authors to uncouple the idea of "feasibility" from this study, as it is not in-tune with the current state of this topic. Sure, "larger trials" are always needed, but that doesn't relegate smaller ones to being simply for "feasibility".

The feasibility aspect of this trial was to assess the feasibility of delivering a future trial rather than the feasibility of the intervention working. Again this is highlighted and discussed in our previous report.

2. The authors need to better clarify their recruitment numbers. The eligibility for this study appears to allow flexibility in recruitment (compared to many published trials which have more stringent criteria for duration of symptoms, WBC, and appendiceal size). This appears to have been done intentionally to allow for "replication of practice patterns" and delivery of a "pragmatic" trial as described by the authors. That is quite reasonable, but with such loose inclusion criteria, it creates the potential for incredible bias in patient selection, in that we don't have the denominator as to how many patients were truly eligible. It appears that only 114 patients were approached to enroll (authors report 57 enrolled corresponding to a 50% enrollment rate). I find it hard to believe that across 3 large UK centers over 12 months, only 114 patients presented with uncomplicated appendicitis, meaning that many more were likely eligible but not approached for enrollment... why? And the "why" is actually less important than just reporting what the total eligibility was. Indeed, recruitment into these studies is incredibly challenging. In fact, 50% enrollment is extremely high compared to published studies (our enrollment rate was only 16%) -- thus, it suggests ALL eligible patients were likely not approached, and thus selection bias looms large. I would recommend inclusion of a flow chart of patients which would provide clarity here, starting at eligibility (something like Figure 1 in our referenced paper).

These details are relevant to our previous report in which they are included.

3. The reported complication rate for the operative appendicitis cases seems unusually high (30%) for uncomplicated appendicitis, inclusive of multiple readmissions. The authors should provide more details on what these complications were. This certainly will make the nonoperative treatment pathway seem more favorable against an elevated surgical complication rate. Again, this may be the result of "non-stringent" inclusion criteria. It appears that 30% of the presumably uncomplicated appendicitis patients in the operative group actually had perforated appendicitis! These patients never should have been enrolled, and certainly would drive the complication rate higher in an otherwise uncomplicated cohort. This further speaks to the sub-

optimal inclusion/exclusion strategy. The authors state diagnostic imaging is not always utilized -- I would counter and say that if you are considering nonoperative management, you want to be as certain as possible that this truly is uncomplicated appendicitis -- and imaging can certainly help that -- otherwise you are setting the antibiotics-only patients up to fail.

These details are relevant to our previous report in which they are included.

4. Similarly, the true success rate of the nonoperative arm was only 44%. The authors report 70% in the nonop group were discharged home, but then 9 were readmitted (7 w/ recurrent appendicitis) -- that math yields 44% success, which is quite low -- now several who failed (n=4 i think) were also found to have perforated appendicitis, again revealing that these patients never should have been included in the study.

5. While the authors' findings of the improved patient-centered outcomes in the nonop group are consistent with other published studies, post-discharge data were only available in 26/57 participants -- this is less than 50%, and likely insufficient to provide valid conclusions on the patient-centered outcomes.

These limitations are not necessarily disqualifying for publication, but need to be proactively addressed in the methods and discussion sections.

We agree that the small sample size is a limitation and have expressed this at the very beginning of the discussion in a revised submission.

Reviewer: 4

Dr. Erik Skarsgard, BC Children's Hospital Comments to the Author This manuscript is from a feasibility RCT comparing outcomes of non-operative treatment vs appendectomy for simple acute appendicitis in children. The trial's primary outcome (recruitment rate), as well as details of secondary, 6-month clinical outcomes have been published previously (Hall NJ, Eaton S, Sherratt FC, et al. CONservative TRreatment of Appendicitis in Children: a randomised controlled feasibility Trial (CONTRACT). Arch Dis Child. 2021 Jan 13;106(8):764–73). These outcomes include initial success rate of non-operative treatment, rate of recurrent appendicitis and overall rate of avoidance of appendectomy in the non-operative group; and in the appendectomy group, histologic findings in the resected appendix, and postoperative complications and readmissions. This manuscript also includes a detailed tabulation of adverse events, severity assignment and actions taken for both groups.

The current manuscript cites this publication (reference 2) and acknowledges the previous reporting of recruitment rate "and other feasibility outcomes". In the Methods of the current manuscript, a bulleted list of outcomes is provided under the Heading of "Secondary Trial Outcomes reported here...:". These include "safety and overall success of initial non-operative treatment", "complications of disease and treatment" and "rate of recurrent appendicitis during 6-month followup period". Although not explicitly stated, it is implied that these outcomes have not been previously reported, however, after reading reference 2, I believe that these outcomes have been published. The remaining outcomes (length of stay, profile of discharge recovery and health-related QOL are new.

While I accept that it is reasonable, and often "good science" to publish multiple articles from a single study dataset, it is the authors' responsibility to ensure avoidance of redundancy in these publications, or if it is unavoidable, to explicitly acknowledge it--and I don't think that has happened here.

We apologise if it appears that we were seeking dual publication. Rather we wished to report additional outcomes and provided greater detail of the whole trial for context. We have now limited this shorter report really to the bare minimum in terms of methods and only reported 'new' results.

The remainder of my comments are in reference to the outcomes that have not been previously reported.

1. Length of hospital stay is reported in table 2 and includes time measurements from randomization to decision to discharge and actual discharge. No explanation is provided to explain the process from decision to actual discharge, but it is presumed that this would be agnostic to the treatment arm. Unless some additional explanation is provided, I'd suggest that only time to actual discharge be presented, since this is the outcome that matters from the patient/family perspective.

No longer relevant in revised shorter report

2. The outcomes reported by the patient diary cards include duration of analgesia use, return to normal/full activities and parental absence from work, and are based on the return of these cards by families. Not only was the overall rate of return low, it was also disparate between groups (11/29=38% in the non-operative group and 16/28 = 57% in the operative group). I think that the results need to be interpreted in the context of the low and unequal respondent rates of the groups. Other than the general acknowledgement of study limitations of a small sample size and short duration of followup, there is no mention of the possibility of contribution of sampling bias in the family's decision to complete or return the diary cards. I accept that this is a "feasibility" trial which justifies the small sample size, but I still think recognition and discussion of these challenges is important, since they will require mitigation in the full trial.

We agree that the small sample size is a limitation and have expressed this at the very beginning of the discussion in a revised submission.

3. Quality of life was assessed at multiple time points using the CHU-9D tool. No description of the tool itself, to whom or how it was administered (patient or parent), and evidence of its previous validation in childhood appendicitis populations is provided. This should be included in the methods.

We believe the CHU-9D is a well recognised and validated tool for HRQoL assessment in children. We have included a reference to this.

Reviewer: 5

Mrs. Coral Smith, University of Nottingham School of Medicine Comments to the Author I really enjoyed the content of your publication, It was an interesting article which contributed to an under researched area. These are my comments for your consideration. Page 2, the last sentence of the background under abstract is too long without punctuation. Page 3 line 25 consider a comma after surgeons. First use of RCT I cant see you spelt it out (not sure if you need to as a well recognized abbreviation). Page 3, Under introduction line 31/32 cross off one of the "ins". Page 3 under Methods line 49/50 consider a comma after March 2017. Page 5, non operative treatment arm - line 5 consider changing "are" to were. and I naturally put a pause after the word operation in the same sentence so consider a comma there. Lastly page 14, line 50 measuring is spelt wrong.

Thank you for suggesting these improvements to writing. Given our complete revamp of this manuscript, most of these suggestions have now become redundant. However, we very much appreciate your help.